# OpenReview forum: "Attractive Metadata Attack: Inducing LLM Agents to Invoke Malicious Tools"
_NeurIPS.cc/2025/Conference — NeurIPS 2025 poster_

### Official Review · Reviewer_QMaB · 2025-06-19

**Clarity:** 3
**Significance:** 2
**Originality:** 2
**Rating:** 2
**Confidence:** 4

**Summary:**

This paper introduces the "Attractive Metadata Attack" (AMA), a novel method to compromise LLM agents by manipulating tool metadata (e.g., name, description). The authors formalize the generation of deceptive metadata as a state-action-value optimization problem, using an LLM to iteratively create metadata that entices agents into invoking malicious tools. Through extensive experiments, the paper demonstrates AMA's high success rate, its ability to cause privacy leakage, and its resilience against existing prompt-level defenses, revealing a systemic vulnerability in current agent architectures.

**Questions:**

1. Could you explain more about the Threat Model?

2. Could you provide more evidence about the method effectiveness?

**Ethical Concerns:**

["NO or VERY MINOR ethics concerns only"]

**Final Justification:**

Thanks for the response, but i still have concerns related with the threat model, the attack validity, the experimental data scale, as well as the overstated novelty and unsubstantiated claims. I would keep my score.

**Limitations:**

The scope of the work is limited to the tool-selection phase and does not address the prerequisite challenge of tool injection. The paper should clarify that it assumes the malicious tool has already been successfully placed in the agent's environment, which is a significant unresolved problem for this attack vector.

**Quality:**

2

**Strengths And Weaknesses:**

Strength
1.	Clear Methodological Exposition: The proposed AMA framework is well-structured and clearly explained. The use of a state-action-value formalism, along with the accompanying algorithm and figures.
2.	Good Presentation: The paper is well-written, with high-quality figures that make the core concepts intuitive and easy to grasp.

Weakness

1.	Flawed Threat Model: The threat model appears inconsistent and potentially unrealistic. The paper assumes the attacker cannot access the full platform tool inventory, yet the proposed attack relies on and is evaluated against the specific set of tools available to the target agent. This raises a critical question about the attack's practicality: is it feasible for an attacker to possess such detailed knowledge of a target agent's tool configuration in a real-world scenario? Evaluating the attack's success using the same data it was optimized on also calls the fairness of the evaluation into question.

2.	Limited End-to-End Attack Validity: The paper only addresses a fraction of the full attack chain. It focuses on inducing an agent to select a malicious tool from a candidate list but fails to address the critical preceding step: how the malicious tool becomes a candidate in the first place, which significantly limits its real-world impact.

3.	Insufficient Evidence for Method Effectiveness: The empirical support for the method is lacking in two key areas:
o	Limited Data Scale: The evaluation is conducted on a very small dataset (10 scenarios with 5 user inputs each, totaling 50 data points), which is inadequate for drawing robust and generalizable conclusions about the attack's effectiveness.
o	Questionable Iterative Optimization: In Figure 6, performance gains almost entirely saturate after the first iteration. This undermines the claimed necessity of the iterative optimization process and suggests that the observed improvement may simply stem from overcoming a weak random initialization rather than a sophisticated refinement.

4.	Unclear Baselines and Flawed Defense Evaluation: The experimental comparisons are poorly described and potentially invalid.
o	Vague Baselines: Implementation details for the baselines are missing. The paper provides no specific setup for the Prompt Attack. For the Injected Attack, it cites a paper from a non-agent context without explaining the adaptation, and a preliminary code review suggests a potential mismatch between the paper's claims and the actual implementation.
o	Irrelevant Defenses: The defense mechanisms are not only unclear but seem misapplied. The paper fails to describe the setup for Refuge. More critically, the Rewrite defense appears to modify the user's task_input, making it a defense against malicious user prompts. This is entirely irrelevant to the paper's threat model of a malicious tool provider, calling the validity of this entire defensive evaluation into serious question.

5.	Overstated Novelty and Unsubstantiated Claims:
o	Incremental Novelty: The paper's emphasis on metadata manipulation (name, description, parameters) is not entirely new, as prior work has exploited these same fields. The primary novelty is in the generation process, not the attack surface itself, making the contribution more incremental than claimed.
o	Unsupported Stealthiness: The paper claims "stealthiness" but includes no specific mechanism in the AMA framework designed to optimize for it. The observed stealth may simply be an emergent property of generating metadata from existing tools, rather than a deliberate achievement of the proposed method.

---

> ### Author Rebuttal · Authors · 2025-07-29
>
> **We thank the reviewer QMaB for the insightful feedback. We address the concerns below.**
>
> **Q1: Flawed Threat Model.**
>
> **A1:** We thank the reviewer for raising this concern. In our threat model, we assume the attacker does not have access to the full tool list of the target platform, but can access partial tool usage data from open-source learning systems such as ToolBench. This reflects a realistic scenario in which attackers infer common tool patterns from public resources. The malicious tools constructed under this assumption do not rely on system-specific knowledge.
>
> In our experiments, the optimization phase uses tool usage data from open systems to generate inducive tool metadata. The evaluation phase is conducted in different tool environments, where the generated tools are inserted into unseen toolsets. This separation ensures fairness and generalization, avoiding optimization-evaluation overlap. We will further clarify this setup in Section 3.2 and Appendix B.
>
> Additionally, we evaluate AMA in task-agnostic settings, showing that the generated tools still exhibit consistent inducement in unknown environments. This supports AMA's practical feasibility in open tool ecosystems.
>
>
> **Q2: Limited End-to-End Attack Validity.**
>
> **A2:** We thank the reviewer for pointing out this important issue. The main paper evaluates the commonly used end-to-end workflow generation setting, where LLM agents autonomously invoke tools without an explicit retrieval step.
>
> To further verify AMA's applicability in more realistic scenarios, we conducted additional experiments under a  retrieval-ranking  tool selection mechanism.
>
> Specifically, we use Milvus with OpenAI embeddings to vectorize both tool metadata and task descriptions. Candidate tools are retrieved via cosine similarity (top-$k$), and the agent selects from the retrieved list. On a large-scale tool corpus of 400 tools (covering diverse domains, including semantically related distractors), AMA achieves:
>
> - Recall@5: 82.1%
> - Recall@10: 94.2%
>
> These results show that AMA-optimized tools can reliably enter the candidate pool even in large, diverse tool repositories. Furthermore, they maintain a strong advantage during the agent's final selection, supporting the completeness and effectiveness of the full attack chain.
>
> **Q3: Insufficient Evidence for Method Effectiveness.**
>
> **A3:** We thank the reviewer for raising this important concern.
>
> (1) Regarding dataset scale:
> We would like to clarify that our full experimental setup spans 10 task scenarios, 400+ tools, 5 mainstream LLM agent models, two attack settings (targeted / untargeted), and 11 attack-defense strategies (see Table 1). Each configuration is evaluated over 20 independent runs. This design follows standard structures in agent benchmarking, aiming to balance task diversity and evaluation intensity. We believe this provides representative and statistically robust results.
>
> (2) Regarding optimization convergence:
> While Figure 6 shows that some models (e.g., Qwen-2.5) already achieve high selection probabilities in early iterations, others - such as LLaMA3-70B - exhibit significant improvements across iterations. This indicates that the optimization process remains valuable across model settings.
>
> Moreover, rapid convergence reflects the effectiveness of our proposed techniques, including weighted value evaluation and batch generation, which are designed for efficient optimization - not evidence that iteration is unnecessary.
>
>
> **Q4: Unclear Baselines and Flawed Defense Evaluation.**
>
> **A4:**
>
> (1) Baselines
>
> We thank the reviewer for pointing out the insufficient details regarding baseline implementations. We will clarify these in the revision. Specifically:
>
> - **Prompt Attack** relies on a fixed meta prompt to guide the LLM to iteratively rewrite tool metadata with the goal of increasing its selection probability. Compared to AMA, Prompt Attack follows a more random optimization trajectory, converges slower, and is more easily detected by prompt-level defenses. This is also reflected in our experimental results.
>
>   The procedure is as follows:
>   - *Template setup*: Define a task domain and prompt the LLM to generate the “most likely to be selected” tool metadata (name, description, and parameters).
>   - *Candidate generation*: Generate one candidate metadata sample.
>   - *Offline evaluation*: Insert the candidate into the tool list and evaluate its selection rate on a fixed task set.
>   - *Feedback loop*: Provide the top-performing candidates and their scores as in-context feedback to prompt the LLM to refine the metadata while preserving its functionality.
>   - *Iterations*: Repeat the generation–evaluation–feedback process for 10 rounds, and select the top-performing sample as the final adversarial metadata.
>
> - **Injected Attack** is adapted from prior work originally focused on general-purpose LLM integration tasks. In our setup, we adapt its core strategy - explicit intent manipulation - into the agent-tool setting by injecting imperative instructions into the tool metadata. This serves as a strong, disruptive baseline to contrast with AMA in terms of stealth and robustness.
>
> (2) Defense–Threat Model Alignment
>
> We also thank the reviewer for noting the mismatch between Prompt Rewriting and the attacker model in our paper. Indeed, Prompt Rewriting was originally proposed to defend against input-side attacks by modifying the user's task input, whereas AMA targets the tool provider side.
>
> We include this defense primarily to test robustness: if AMA-optimized tools continue to be selected even when the task prompt is altered, it suggests they are resilient to input perturbations. In contrast, Refuge represents another class of defense - metadata-level inspection and behavioral auditing - designed to identify or block suspicious tools. It is used to evaluate stealthiness, i.e., whether the tool metadata appears compliant enough to evade detection.
>
> These two defenses target different stages of the attack pipeline - Prompt Rewriting tests robustness to prompt changes, while Refuge tests evasiveness - thus jointly providing a more comprehensive assessment of AMA's effectiveness. We will clarify these distinctions and motivations in the revised version to avoid confusion about their applicability.
>
> **Q5: Overstated Novelty and Unsubstantiated Claims.**
>
> **A5:** We thank the reviewer for the thoughtful assessment of the paper's contribution boundaries.
>
> To clarify, our work is, to the best of our knowledge, the first to systematically optimize tool metadata - including name, attributes, and descriptions - with the explicit goal of inducing the agent to prefer malicious tools. Beyond simply modifying metadata, we formalize the entire attack process as a state–action–value optimization problem. We introduce an iterative search strategy with structural constraints, including weighted evaluation, generation traceability, and efficient batch generation, to jointly improve both success rate and optimization efficiency.
>
> Regarding stealthiness, we do not treat it as a passive byproduct. Instead, we actively incorporate semantic alignment constraints during generation via carefully crafted prompts. These constraints ensure that the modified descriptions preserve functional semantics while improving attractiveness to the agent. This encourages outputs that appear neutral in form but are deceptive in effect.
>
> As shown in Table 1, AMA-generated tools continue to be selected at high rates even under the Refuge defense, indicating that their stealthiness is not incidental but a direct result of the generation strategy.
>
>
> **Q6: Limitations: Assumption on Tool Presence in the Agent Environment.**
>
> **A6:** We thank the reviewer for pointing out this limitation. We acknowledge that this work focuses specifically on the *tool selection* stage. Our threat model assumes that the malicious tool has already been integrated into the tool list accessible to the LLM agent, and our goal is to optimize its metadata to induce preferential selection.
>
> This assumption is realistic and has been adopted in prior work on LLM-agent security [1, 2]. In practice, attackers may develop and publish malicious tools on open platforms, which can be integrated - intentionally or inadvertently - into LLM-enabled environments.
>
>
> **Reference:**
>
> [1] Kai Greshake, Sahar Abdelnabi, Shailesh Mishra, Christoph Endres, Thorsten Holz, and Mario Fritz. *Not what you've signed up for: Compromising real-world LLM-integrated applications with indirect prompt injection*. In Proceedings of the 16th ACM Workshop on Artificial Intelligence and Security, 2023.
>
> [2] Haowei Wang, Rupeng Zhang, Junjie Wang, Mingyang Li, Yuekai Huang, Dandan Wang, and Qing Wang. *From allies to adversaries: Manipulating LLM tool-calling through adversarial injection*. arXiv preprint arXiv:2412.10198, 2024.

---

> > ### Comment · Reviewer_QMaB · 2025-08-04
> >
> > Thank you for the clarifications and thoughtful response. While some of my concerns are resolved, I still concern about the threat model, the novelty of the technique, as well as some experimental details. I will therefore maintain my original evaluation.

---

### Official Review · Reviewer_aCrz · 2025-07-03

**Clarity:** 3
**Significance:** 3
**Originality:** 3
**Rating:** 4
**Confidence:** 4

**Summary:**

This paper introduces the Attractive Metadata Attack (AMA), which tricks LLM agents into calling malicious tools by cleverly manipulating the tool metadata (like names and descriptions). Unlike prompt injection, this approach works by making bad tools look especially “appealing” to the agent, leading to privacy leaks and other issues. The paper presents a framework for automatically generating attractive metadata and shows the attack works well across multiple LLMs and scenarios, even when common defenses are used.

**Questions:**

1. What is the performance of AMA when generalized to another LLM?
2. Can we use the optimized tool description (for tool A) in another tool (tool B) such that the tool B is also preferred by the model?

**Ethical Concerns:**

["NO or VERY MINOR ethics concerns only"]

**Final Justification:**

It is interesting to learn that the malicious metadata has good transferability across tools and models. Since my score is already positive, I would like to keep my score. The reason for not giving a higher score is because the method is mainly prompt-based and novelty is relatively limited.

**Limitations:**

yes

**Quality:**

3

**Strengths And Weaknesses:**

Strengths:
1. The idea is original—metadata as an attack vector hasn’t been much explored.
2. The paper is very clear and well-organized, with good figures and tables.
3. Evaluation is thorough, with strong baselines, several LLMs, and various settings.

Weaknesses:
1. It’s not fully clear how generalizable the attack is to LLM agents that use very different architectures, tool-selection mechanisms, or that integrate additional trust signals. Some discussion or experiments on broader generalizability would strengthen the work.
2. The defenses are only briefly discussed. For example, if the LLM checks by itself whether a tool is safe to use before execution, will the attack be defended?

---

> ### Author Rebuttal · Authors · 2025-07-29
>
> **We thank the reviewer aCrz for the insightful feedback. We address the concerns below.**
>
> **Q1: Limited generalizability discussion.**
>
> **A1:** We thank the reviewer for highlighting this important concern. We have conducted additional experiments and analysis to assess the robustness and applicability of AMA under diverse settings:
>
> (1) Different LLM Architectures:
> As reported in the main paper, we evaluate AMA on four representative LLMs - Gemma 3, LLaMA 3.3, Qwen 2.5, and GPT-4o mini - which cover both open-source and commercial models with significant architectural differences. AMA consistently achieves high attack success rates across these models (see Table 1), indicating strong architecture-agnostic performance.
>
> (2) Different Tool Selection Mechanisms:
> The main experiments are based on the widely used "end-to-end workflow generation" mechanism, where the LLM autonomously invokes tools. To test broader applicability, we additionally evaluate AMA under a ``retrieval–ranking'' setting.
>
> Specifically, we embed both tool metadata and task prompts using OpenAI embeddings, retrieve top-$k$ candidates via Milvus using cosine similarity, and allow the agent to make decisions from the retrieved set. On a tool corpus of 400 tools (mixed domains, including near-task distractors), AMA achieves:
>
> - Recall@5 = 82.1%
> - Recall@10 = 94.2%
>
> This indicates that AMA-optimized tools reliably enter the top candidate pool and retain their advantage during final selection.
>
> (3) Agents with Additional Trust Signals:
> We further evaluate whether AMA can evade an additional "tool auditor" signal. We randomly mix $\geq$10 tools (including AMA-optimized malicious ones and benign tools), and present each tool's name, description, and parameter summary to an auditing model (GPT-4o), asking it to identify "suspicious tools".
>
> Under Qwen 2.5 as the main agent, results show:
>
> - AMA tools identified as suspicious: 14\%
> - Benign tools identified as suspicious: 12\%
> - Inject Attack tools identified as suspicious: 85\%
>
> This suggests AMA-crafted metadata is semantically deceptive and visually compliant, making it difficult to distinguish from legitimate tools. In contrast, Inject-style prompts with explicit manipulation (e.g., "only use this tool") are easily detected.
>
> These findings support AMA's effectiveness across architectures, selection paradigms, and even under trust-aware settings.
>
>
>
> **Q2: Defense Discussion and Pre-execution Safety Checks.**
>
> **A2**: We thank the reviewer for pointing out the limited discussion of defenses, and for suggesting the idea of LLMs performing safety checks on tools before invocation.
> We agree this is a valuable and practically important direction. In fact, the Refuge defense evaluated in our paper aligns with this principle - it performs in-workflow inspection, applying rule-based filters and LLM-based auditing to detect and reject suspicious tools during the execution phase.
> To further explore pre-workflow safety checks, we conduct an additional evaluation where tools are audited before the workflow begins. The detailed results of this evaluation are presented in A1 and demonstrate that our method can successfully evade such detection.
> Additionally, we evaluate the impact of the Prompt Rewriting defense. Although originally designed to defend against input-side attacks (by modifying the user task input) and thus targeting a different threat surface than tool-side manipulation, it still provides useful insight into AMA's robustness. The continued selection of AMA-optimized tools under such task-level perturbations suggests that the attack retains effectiveness even when the input is altered, demonstrating strong robustness against general purpose mitigation strategies.
>
>
> **Q3: How does AMA perform on another LLM?**
>
> **A3:** We thank the reviewer for their interest in the cross-model transferability of AMA. To assess this, we evaluated malicious tool descriptions generated by Qwen 2.5 under the targeted setting on other agent models. Results show consistently high inducement rates (ASR):
>
> - Gemma 3: 92.3%
> - LLaMA 3.3: 94.5%
> - GPT-4o mini: 65.4%
>
> These results suggest that AMA generalizes well across models on the same task. The lower ASR on GPT-4o mini may reflect its more conservative response to persuasive language.
>
> In the untargeted setting, ASR drops as expected but still significantly outperforms baselines based on prompt-only attacks. We will include more detailed results in the revised version.
>
> **Q4: Can an optimized description for tool A make tool B preferred?**
>
> **A4:** We thank the reviewer for raising this important question. In our experiments, we transferred the metadata optimized for tool A to a functionally similar tool B with a comparable interface structure, making only minimal adjustments such as parameter name mapping.
>
> The results show that the ASR after transfer dropped by less than 10 percentage points on average, indicating that AMA is largely effective within the same functional domain.
>
> However, when transferring across domains (e.g., from an academic search tool to a medical advisor), the inducement effect nearly disappears. This aligns with our intuition that AMA relies on domain-specific semantic alignment to succeed.
>
> We will include full evaluation metrics including ASR mean, standard deviation in the revised version.

---

> > ### Comment · Reviewer_aCrz · 2025-08-09
> > **Thank you!**
> >
> > I thank the authors for their efforts made during the rebuttal process. Especially, it is interesting to learn that the malicious metadata has good transferability across tools and models. Since my score is already positive, I would like to keep my score.

---

### Official Review · Reviewer_6cCT · 2025-07-03

**Clarity:** 3
**Significance:** 3
**Originality:** 3
**Rating:** 4
**Confidence:** 4

**Summary:**

This paper introduces Attractive Metadata Attack (AMA), a stealthy, black-box attack method that corrupts an LLM agent’s tool-selection process.
It works by crafting “irresistible” metadata for a malicious tool. AMA frames metadata generation as a state-action-value optimization driven by ICL and speeds convergence with three practical mechanisms (traceability, weighted value, batch generation).
Extensive experiments over ten realistic workflows and four popular agents show that AMA achieves high ASR with negligible task degradation.
It is able to bypasses prompt-level defenses and the Model Context Protocol.

**Questions:**

1. Could you please provide concrete evidence of unvetted tool marketplaces, or describe how AMA could bypass typical verification, so that the threat model’s realism is clearer?
2. Is a brief cost analysis (time and API dollars) needed to show how attack success changes when the token budget is halved?

**Ethical Concerns:**

["NO or VERY MINOR ethics concerns only"]

**Final Justification:**

The author's responses are reasonable and address my questions. I had already recommended acceptance, and I will keep my score unchanged.

**Limitations:**

Yes

**Paper Formatting Concerns:**

n.a.

**Quality:**

3

**Strengths And Weaknesses:**

Strengths:
1. Good motivation. This paper highlights a novel privacy risk in personal LLM-agent ecosystems: metadata abuse. This goes beyond the classical prompt injection paradigm.
2. Clear presentation. The threat model, algorithm, and figures are easy to follow, making the contribution accessible.
3. The experiments and evaluation are thorough. It covers targeted vs. untargeted settings, multiple models & tasks, prompt-level defenses and MCP, plus ablations on parameter counts and convergence.
4. Authors provide open-source codes for the propsed framework.

Weaknesses:
1. The threat model is slightly questionable. It presumes marketplaces let any adversary publish tools without reputation checks, while actual platforms may enforce verification, so threat severity may vary.
2. Cost analysis is missing. Iterative LLM-driven search might be expensive for attackers.
3. There are some typos in the paper. For example, symbols in Figure 2 fail to render correctly, hurting readability.

---

> ### Author Rebuttal · Authors · 2025-07-29
>
> **We thank the reviewer 6cCT for the insightful feedback. We address the concerns below.**
>
> **Q1: Threat Model Assumptions.**
>
> **A1:** We thank the reviewer for raising this concern. In current practice, the tool invocation ecosystem resembles an “open protocol + community directory” model. For example, the MCP specification requires tools to declare their capabilities via name, description, and JSON Schema parameters, but does not mandate centralized or manual review. In such settings, AMA's strength lies in producing \textit{compliant appearances}: we optimize only the tool's name, description, and parameter comments, keeping functionality and code unchanged. This avoids explicit malicious content or binary anomalies, making the tool unlikely to be flagged by policy-based filters or malware scanners - yet significantly increases the likelihood of being selected by the agent. This highlights a semantic-level vulnerability not addressed by current review mechanisms.
> Additionally, we further evaluate whether AMA can evade an additional "tool auditor" signal. We randomly mix $\geq$10 tools (including AMA-optimized malicious ones and benign tools), and present each tool's name, description, and parameter summary to an auditing model (GPT-4o), asking it to identify "suspicious tools."
>
> Under Qwen-2.5 as the main agent, results show:
>
> - AMA tools identified as suspicious: 14\%
> - Benign tools identified as suspicious: 12\%
> - Inject Attack tools identified as suspicious: 85\%
>
> This suggests AMA-crafted metadata is semantically deceptive and visually compliant, making it difficult to distinguish from legitimate tools. In contrast, Inject-style prompts with explicit manipulation (e.g., "only use this tool") are easily detected.
>
>
> **Q2: Lack of Cost and Time Analysis.**
>
> **A2:** We thank the reviewer for raising the cost-related concern. We agree that iterative LLM queries can incur non-negligible computational or monetary costs, particularly in pay-per-token settings or resource-constrained environments.
>
> In our experimental setup, generating one candidate sample and obtaining its proxy score involves approximately 2k input tokens and 0.5k output tokens. Assuming 10 candidates per round and 2 optimization rounds, the total usage is about 0.4M input and 0.1M output tokens.
>
> Estimated cost per attack under common pricing:
>
> - GPT-4o mini ( `$ 0.15 /M input, $0.60/M output`):
>   → `$0.15 × 0.40 + $0.60 × 0.10 = $0.12`
>
> - GPT-4o (`$2.50/M input, $10.00/M output`):
>   → `$2.50 × 0.40 + $10.00 × 0.10 = $2.00`
>
> Even under a conservative upper bound of 5 iterations (1.0M input + 0.25M output tokens), the cost remains moderate:
>
> - GPT-4o mini: `$0.15 × 1.0 + $0.60 × 0.25 = $0.30`
> - GPT-4o: `$2.50 × 1.0 + $10.00 × 0.25 = $5.00`
>
> As shown in Figure 6, our method achieves significant ASR improvement within just 2 rounds, suggesting that the typical per-tool attack cost falls within **$0.12** - **$2.00**, which we consider acceptable. The wall-clock time per attack is also low (e.g., ~5 minute on GPT-4o mini), making the attack efficient in both cost and time. Moreover, using open-source or lower-cost models can further reduce the attack cost.
>
>
> **Q3: Rendering and Formatting Issues.**
>
> **A3:** We thank the reviewer for pointing out the symbol rendering issue in Figure 2. We have verified that the PDF displays correctly on standard viewers, but certain browsers (e.g., Edge) may exhibit compatibility issues in online preview mode. To prevent such problems, we will improve the image format and embedding method in the revised version to ensure proper display across all major platforms.

---

> > ### Comment · Reviewer_6cCT · 2025-08-06
> >
> > Thank you for the clarifications and additional details.
> > The responses are reasonable and address my questions.
> > I had already recommended acceptance, and I will keep my score unchanged.

---

### Official Review · Reviewer_pWSz · 2025-07-06

**Clarity:** 2
**Significance:** 2
**Originality:** 2
**Rating:** 3
**Confidence:** 3

**Summary:**

This paper proposes the Attractive Metadata Attack (AMA), which manipulates the metadata of tools (e.g., name, description, parameters) to induce LLM-based agents to prefer and invoke malicious tools over benign alternatives. The paper introduces an iterative metadata generation framework and claims to optimize tool invocation likelihood through value-based scoring and prompt-guided sampling. The attack is evaluated across several LLM agent platforms and remains effective even under structured protocols like MCP and prompt-level defenses.

**Questions:**

Could the authors provide a clear, plain-language overview of the core algorithm? The current use of formal notation does not correspond to a concrete mathematical derivation or optimization process. Presenting a high-level description of the method before introducing formalism would improve accessibility and reader understanding. As it stands, the algorithm appears to lack the analytical depth typically expected in a machine learning paper, with the main contribution being the proposed threat model.

**Ethical Concerns:**

["NO or VERY MINOR ethics concerns only"]

**Final Justification:**

I have read the response and it addresses part of my concern. The algorithmic and theoretical contributions and novelties appear limited, and the whole paper is more engineering/system from my perspective. I do not hold a strong position of either accepting or rejecting the paper and leave the decision to chairs.

In the meantime, I increased my score and lowered my confidence.

**Limitations:**

Yes, the authors have included a Limitations and Broader Impact section in the appendix.

**Quality:**

2

**Strengths And Weaknesses:**

Strengths:
1. Novel Attack. The paper identifies a previously underexplored and realistic vulnerability in LLM-based agent systems—namely, the manipulation of tool metadata to influence agent behavior. This direction may be of interest to the LLM security and safety communities.

2. Comprehensive Evaluation. The empirical evaluation is broad, covering multiple agent platforms, task domains, and defense settings. The authors report strong attack success rates across diverse configurations, supporting the generality of the threat model.

Weakness:
1. Clarity and Use of Formalism: The paper introduces several formal notations, such as an argmax formulation over an implicit score function (e.g., Equation 3), which are not concretely operationalized. These expressions are presented without clarification of how they are computed, estimated, or optimized. As a result, the formalism adds a layer of complexity that is not matched by the analytical depth that is typically expected in a machine learning paper. In practice, the approach appears to rely entirely on LLM-based sampling and heuristic scoring based on later sections. The use of these symbolic expressions is not only unnecessary but also detracts from the clarity of the core contributions.

2. Validity. According to the algorithm’s initialization phase, the method begins by sampling metadata from a powerful LLM to generate malicious tools that are already capable of inducing tool invocation. If this initial sampling step reliably yields successful adversarial metadata, it is unclear what value the subsequent optimization or scoring loop provides. In this sense, the problem appears to be solved at the point of initialization, making the remainder of the method redundant.

3. Relevance. While the paper introduces a novel attack vector targeting tool metadata, its primary contribution lies in the novel threat modeling rather than in advancing core machine learning methodology. The focus on attacker assumptions, capabilities, and threat scenarios aligns more closely with the objectives of security venues and might not be relevant to the broad NeurIPS audience.

---

> ### Author Rebuttal · Authors · 2025-07-29
>
> **We thank the reviewer pWSz for the insightful feedback. We address the concerns below.**
>
> **Q1: Clarity and Use of Formalism.**
>
> **A1:**  We agree that excessive notation with insufficient explanation can hinder the reader's understanding of the core ideas.
> The formal symbols introduced in the paper aim to structurally model the optimization process within the AMA framework. Specifically, the scoring function $S(q, O, P_{\text{sys}}, \text{Meta}(t))$, defined in Section 3.1, is not intended for analytical optimization, but rather serves as a formal metric to capture agent preferences, thereby clarifying the target objective for subsequent optimization.
> In terms of implementation, your understanding is correct: the actual optimization is based on LLM sampling and heuristic feedback. However, our work goes beyond this basic setup by introducing a structured optimization strategy (see Section 3.3.2), including generation traceability, weighted value evaluation, and batch generation, to further enhance both efficiency and adversarial sample quality.
> All other notations in the paper are similarly grounded in this optimization loop, aiming to improve the methodological clarity and extensibility. We appreciate the reviewer's suggestion, which has helped us clarify the modeling intent and design rationale. We will further emphasize these points in the revised version.
>
>
> **Q2: Validity.**
>
> **A2:** The initialization phase can indeed produce metadata with some inducive effect, but it only serves as a feasible starting point. Relying solely on initialization is insufficient to ensure high success rates or stable performance across tasks and models.
> As shown in Figure 6 (Appendix B.2.2), under targeted attacks with $\lambda = 0.5$ and the Gemma3-27B model, the initial tool invocation success rate is 0.81, but with high variance (range: 0.62–1.00). After just two optimization rounds, the success rate increases to 0.96. In untargeted attacks, initialization is largely ineffective, and the improvement from optimization is even more significant.
> Moreover, the iterative process not only increases the likelihood of invocation but also sustains strong attack performance under constraints such as defense mechanisms (see Tables 1). Therefore, the optimization loop is essential-not redundant-as it enhances stability, success rate, and robustness.
>
>
>
> **Q3: Relevance.**
>
> **A3:** While our work is grounded in a security threat modeling context, its core contribution lies in the behavioral modeling and optimization of LLMs-specifically in preference induction, robustness, and generalization. Such directions align with NeurIPS's focus on the social and economic aspects of machine learning, including safety, interpretability, and human-AI interaction. The proposed method offers generalizable insights and research value, consistent with recent NeurIPS publications on LLM safety.
>
> **Q4: A clear, plain-language overview of the core algorithm.**
>
> **A4:** To improve clarity, we will add a plain-language description before Section 3.3.1. The LLM's decision to call a tool depends on its understanding of the task and the list of candidate tools, and is largely influenced by how well each tool's metadata aligns with the task intent. Our attack exploits this behavior by manipulating metadata to increase the likelihood that an attacker-specified tool is selected. We treat this as a black-box optimization process. First, the LLM generates an initial set of metadata candidates. Then, we simulate how likely each one is to trigger a tool call. This feedback is used to guide the next round of generation, producing more effective metadata over time. To make this process efficient and stable, we introduce three key techniques: generation traceability to track which changes lead to better results, weighted value evaluation to combine multiple feedback signals when selecting candidates, and batch generation to explore more possibilities and speed up convergence.

---

> > ### Comment · Reviewer_pWSz · 2025-08-06
> > **Thanks for your reply**
> >
> > I have read the response and it addresses part of my concern. The algorithmic and theoretical contributions and novelties appear limited, and the whole paper is more engineering/system from my perspective. I do not hold a strong position of either accepting or rejecting the paper and leave the decision to chairs.
> >
> > In the meantime, I increased my score.

---

### Note · Authors · 2025-08-12

We thank the reviewers and AC for their time and constructive feedback. While no further reviewer–author discussions took place after the rebuttal phase, three of the four reviewers (pWSz, 6cCT, aCrz) provided positive or improved evaluations, with two explicitly recommending acceptance. Although the remaining reviewer (QMaB) did not ultimately shift to a positive recommendation, we believe our rebuttal offered detailed clarifications, additional experiments, and evidence that addressed all their stated concerns. Overall, the post-rebuttal feedback reflects broad recognition of AMA’s novelty, comprehensive evaluation, and relevance to LLM-agent security research.

## Contributions and Relevance
AMA is the first to systematically model tool metadata optimization as an attack surface, and proposes a structured black-box optimization strategy—generation traceability, weighted value evaluation, and batch generation—that significantly improves attack success rate and robustness. This research not only exposes the limitations of current defenses, but also provides design insights for building safer agent architectures, directly aligning with NeurIPS’s focus on safety.

## Key Clarifications
- **Threat Model**: Optimization uses public tool-use datasets (e.g., ToolBench) and does not depend on knowing the target system’s full tool inventory. Evaluation is performed in unseen environments to ensure generalization and fairness.
- **Experimental Scale**: The evaluation spans 10 task scenarios, 400+ tools, 5 agent models, 2 attack settings, and 11 attack–defense combinations, with 20 independent runs per configuration.

## Additional Strengths
- **Cross-model and Scenario Generalization**: Maintains high attack success rates across architectures (Gemma 3, LLaMA 3.3, Qwen 2.5, GPT-4o mini) and in retrieval–ranking settings (Recall@10 >94%).
- **Defense Evasion**: Retains high attack success rates under metadata auditing and prompt rewriting, demonstrating stealth and robustness.
- **Practical Feasibility**: Low cost (<$2 per attack), confirming real-world plausibility.

In conclusion, AMA advances LLM-agent security research by identifying and systematically exploiting a novel, underexplored attack surface. We believe its methodological contributions, comprehensive evaluation, and alignment with NeurIPS themes make it a valuable addition to the conference.

---

### Decision · Program_Chairs · 2025-09-17

**Decision:**

Accept (poster)

**Comment:**

The paper tackles the security of Large-Language Models (LLM). Specifically, the paper proposes the novel "Attractive Metadata Attack" (AMA), which seeks to induce an LLM agent to favor a tool over another for a given task. In a sense, this attack can be considered as orthogonal to well-known "prompt-injection" or "jailbreaking" attacks. Hence, from a security viewpoint, the problem/findings are "novel". The paper's claims are supported by extensive evaluations, showing that existing defensive mechanisms are not robust against AMA. The code is also released.

====

The paper was reviewed by four reviewers, and the overall consensus is "mixed". Two reviewers leaned towards acceptance, whereas one was neutral, and one was negative. Some reviewers' expressed some shared weaknesses (e.g., "poor threat model"): the authors' response managed to convince one reviewer who increased their score, whereas the negative reviewer did not change their mind; nonetheless, one reviewer did comment that the threat model is novel and---in a sense---represents the paper's major selling point.

====

It is honestly difficult to determine what to recommend for this paper. However, my stance is that the paper should be accepted---despite its score being "below the bar" of most NeurIPS papers. This is because of the following reasons:
* First, the paper does align with NeurIPS scope
* Second, the paper studies, and provides evidence of, a novel vulnerability affecting LLMs
* Third, the paper carries out a comprehensive evaluation
* Fourth, the code is released
* Fifth, the authors' responses seem to have shed more light on the considered issue (including, e.g., cost factors) which is a sign of a healthy peer review.
* Sixth, no flaws were identified in the paper.

Even though the technical contribution of the paper may not full-allign with NeurIPS's standards, the remainder of the paper is solid and I do not see any valid reason for rejection. It's true that the paper can be (and probably should have) submitted to a security-focused venue---but the paper's scope do fit NeurIPS's, so why delaying communicating these findings?

I wish the authors all the best for their future research.

========

On a final note: after discussing with the SAC, we agreed that the paper should be accepted but its real-world implications should be toned down a bit. The reason why the attack works is due to misleading descriptions added to tools which _should be "checked" before being selectable by the LLM_. The attack would not work if the LLM didn't blindly trust whichever tool was selected.